# Assessment of Grouped Weighted Quantile Sum Regression for Modeling Chemical Mixtures and Cancer Risk

**DOI:** 10.3390/ijerph18020504

**Published:** 2021-01-09

**Authors:** David C. Wheeler, Salem Rustom, Matthew Carli, Todd P. Whitehead, Mary H. Ward, Catherine Metayer

**Affiliations:** 1Department of Biostatistics, School of Medicine, Virginia Commonwealth University, Richmond, VA 23298-0032, USA; rustoms@mymail.vcu.edu (S.R.); carlimm@mymail.vcu.edu (M.C.); 2Division of Epidemiology/Biostatistics, University of California, Berkeley School of Public Health, Berkeley, CA 94704-7394, USA; ToddPWhitehead@Berkeley.edu (T.P.W.); cmetayer@berkeley.edu (C.M.); 3Occupational and Environmental Epidemiology Branch, Division of Cancer Epidemiology and Genetics, National Cancer Institute, Rockville, MD 20850, USA; wardm@mail.nih.gov

**Keywords:** mixtures, environment, cancer, chemicals

## Abstract

Individuals are exposed to a large number of diverse environmental chemicals simultaneously and the evaluation of multiple chemical exposures is important for identifying cancer risk factors. The measurement of a large number of chemicals (the exposome) in epidemiologic studies is allowing for a more comprehensive assessment of cancer risk factors than was done in earlier studies that focused on only a few chemicals. Empirical evidence from epidemiologic studies shows that chemicals from different chemical classes have different magnitudes and directions of association with cancers. Given increasing data availability, there is a need for the development and assessment of statistical methods to model environmental cancer risk that considers a large number of diverse chemicals with different effects for different chemical classes. The method of grouped weighted quantile sum (GWQS) regression allows for multiple groups of chemicals to be considered in the model such that different magnitudes and directions of associations are possible for each group of chemicals. In this paper, we assessed the ability of GWQS regression to estimate exposure effects for multiple chemical groups and correctly identify important chemicals in each group using a simulation study. We compared the performance of GWQS regression with WQS regression, the least absolute shrinkage and selection operator (lasso), and the group lasso in estimating exposure effects and identifying important chemicals. The simulation study results demonstrate that GWQS is an effective method for modeling exposure to multiple groups of chemicals and compares favorably with other methods used in mixture analysis. As an application, we used GWQS regression in the California Childhood Leukemia Study (CCLS), a population-based case-control study of childhood leukemia in California to estimate exposure effects for many chemical classes while also adjusting for demographic factors. The CCLS analysis found evidence of a positive association between exposure to the herbicide dacthal and an increased risk of childhood leukemia.

## 1. Introduction

More than one million Americans and more than 10 million people worldwide are diagnosed with cancer each year and many of these cases are attributable to environmental risk factors [1,2,3,4,5]. In the United States alone, there are more than 80,000 chemicals on the market, and some are found in a wide array of consumer products [6]. As a result, individuals are exposed to a large number of chemicals simultaneously on a daily basis. There is a rich history of investigators conducting epidemiologic studies that evaluate environmental chemical exposures in relation to cancer incidence or mortality with the objective of identifying environmental determinants of cancer [7,8,9,10,11,12,13]. While these epidemiologic studies have proven valuable for studying potential environmental risk factors for cancers, most have taken a simplifying approach of evaluating environmental chemical exposures one at a time. Traditionally, studies of environmental chemical exposures and cancer have used a single-chemical regression approach that evaluates chemicals independently as risk factors. Some studies have focused on the total summed exposure for one specific chemical group, such as polychlorinated biphenyls (PCBs) [7,9,11]. Other studies of environmental factors and disease risk consider many chemical exposures in one chemical class [14], or consider many chemicals from different classes independently from each other and control for multiple comparisons in so-called environment-wide association studies [15,16]. However, these studies cannot estimate effects for simultaneous exposure to multiple diverse environmental chemicals.

A major limitation with single-chemical analyses is that they are subject to residual confounding due to ignoring the effect of other chemicals when evaluating the effect of one chemical on health. The high degree of correlation that is often present among some chemical concentrations [11,17,18] creates a serious issue with confounding that can bias effect estimates for individual chemicals and lead to incorrect inference about risks. Some studies examine the pairwise correlation coefficients between environmental factors [11,17,19], but do not account for the correlation among factors in statistical models. The lack of statistical independence observed among chemical exposures, as well as other environmental and socioeconomic variables, presents a significant challenge to assessing many exposure effects simultaneously in a traditional regression model. Issues associated with collinearity resulting from including several correlated chemicals in the same regression model include sign reversal in estimated regression coefficients and inflated standard errors for regression coefficients leading to incorrect conclusions about health effects of individual chemicals [20].

Exposures are increasingly being measured for a large number of chemicals in epidemiologic studies to allow for a more exhaustive investigation of environmental cancer risk factors [11,17,18]. Modeling environmental chemicals exposures collectively is in agreement with the vision of the “exposome” that seeks to characterize the totality of environmental exposures for cancer risk [21,22]. However, limitations in methods for evaluating correlated environmental exposures over multiple chemical classes in epidemiologic studies have hindered analyses. Weighted quantile sum (WQS) regression [18,23,24] is a constrained regression approach that was designed to estimate the effect of a mixture of correlated chemicals and identify the individual chemicals most strongly associated with a health outcome while adjusting for risk factors. In WQS, a weight is estimated for each chemical in a weighted index, where the weights are constrained to be between 0 and 1 and sum to 1. A substantial limitation of the WQS regression method is that all chemicals in the weighted chemical index in the model are constrained to have associations with the outcome that are in the same direction as all the other chemicals in the index. This is not a realistic constraint when chemicals in different classes have different directions and magnitude of association with a health outcome. For example, organochlorine compounds such as certain PCB congeners have been found to be positively associated with non-Hodgkin lymphoma (NHL) [7], while certain insecticides have been found to be negatively associated with NHL [8]. Alternatively, it could be possible that PCBs have a greater magnitude of effect than polycyclic aromatic hydrocarbons (PAHs) for a particular health outcome.

Given the multitude of diverse chemicals to which individuals are exposed, approaches to modeling environmental chemical cancer risk that consider a large number of chemicals with different effects for different chemical classes are required. The method of grouped weighted quantile sum (GWQS) regression was first proposed to allow for multiple groups of chemicals to be considered in the model such that different magnitudes and direction of associations are possible for each group [25]. However, the approach has not been systematically evaluated as a mixture analysis method capable of estimating exposure effects for multiple groups of chemicals. In this paper, we assessed the ability of GWQS regression to estimate exposure effects for multiple chemical groups and correctly identify important chemicals in each group using a simulation study. We also compared the performance of GWQS regression with WQS regression, the least absolute shrinkage and selection operator (lasso), and the group lasso in estimating exposure effects and identifying important chemicals. Shrinkage methods such as the lasso have been used previously to model chemical mixture effects because they were designed for handling correlated predictor variables [23,24]. As an application, we used GWQS regression in a case-control study of childhood leukemia in California, the California Childhood Leukemia Study (CCLS), to estimate exposure effects for many chemical classes while also adjusting for demographic factors.

## 2. Materials & Methods

### 2.1. Grouped WQS Regression

We focus on the simultaneous exposure to many diverse environmental chemicals and use statistical methodology that extends weighted quantile sum regression to model disease risk for groups of exposures. Studies have shown that WQS regression is more sensitive and specific in identifying important chemicals risk factors than traditional regression and regularization methods such as lasso, adaptive lasso, and elastic net [23]. Grouped WQS was first proposed to allow for multiple groups of chemicals to be considered in the model such that different magnitudes and direction of associations are possible for each group of chemicals [25]. For example, some types of chemicals may have a positive association with disease while others may have a negative association.

Specifically, GWQS uses data with C components (e.g., chemical exposures) split between j=1,…,J groups with $C_{j}$ components in the jth group. Within each of these J groups, the components are scored into quantiles (e.g., quartiles 0,1,2,3) that can be plausibly combined into an index and are assigned a weight. The index weights in each group are empirically estimated and constrained to be between 0 and 1 and sum to 1, which helps reduce potential issues with collinearity and can reduce dimensionality through zero or near-zero weights. For a binary outcome Y, the general GWQS regression model is
(1)$$\log[P(Y=1)/P(Y=0)]=\beta_{0}+\sum_{j=1}^{J}\beta_{j}\left(\sum_{i=1}^{C_{j}}w_{ji}q_{ji}\right)+z^{\prime }\phi ,$$
where $w_{ji}$ represents the weight for the *ith* chemical component in the *jth* group, *q_ji_* is the quantile of the *ith* chemical in the *jth* group, and the summation $\sum_{i=1}^{C_{j}}w_{ji}q_{ji}$ represents a weighted index for the set of $C_{j}$ chemicals of interest within group j. The vector $z^{\prime }$ is a vector of covariates for which to adjust with regression coefficients in the vector ϕ. While this is a model of the log-odds of disease, different link functions can be used depending on the type of outcome (e.g., continuous or count).

For estimation of the model parameters in Equation (1), b=1,…,B bootstrap samples of the training set are taken and nonlinear optimization is used to find the parameter values that maximize the log likelihood. In each bootstrap sample b, the estimated vector of weights is used to form the index and the significance of each group effect $\beta_{j}$ is evaluated through a test statistic $t_{j}$. The final index weights are estimated from all the bootstrap samples using the test statistics as ${\bar{w}}_{ji}=\sum_{b=1}^{B}w_{jib}\left|t_{jb}\right|/\sum_{b=1}^{B}\left|t_{jb}\right|$, which is the weighted average of the component weights using the test statistics as weights over the bootstrap samples for component i in group j. The final estimated index for each of the *j* chemical groups is then calculated as $GWQS_{j}=\sum_{i=1}^{C_{j}}{\bar{w}}_{ji}q_{ji}$. Final estimates of the group exposure effects and associated statistical significance are obtained in a validation data set by fitting the generalized linear model
(2)$$\log[P(Y=1)/P(Y=0)]=\beta_{0}+\sum_{j=1}^{J}\beta_{j}GWQS_{j}+z^{\prime }\phi .$$

In addition to estimating the chemical mixture exposure effects, the method allows one to identify the important chemicals in the mixture through the empirically estimated weights.

We have implemented the nonlinear optimization using the solnp function from the R package Rsolnp [26,27]. This function uses an augmented Lagrange multiplier method with a sequential quadratic programming interior algorithm. Our implementation is publicly available on The Comprehensive R Archive Network (CRAN) as an R package titled groupWQS to allow users to perform GWQS for their own research [28].

### 2.2. Simulation Study Design

To evaluate the performance of GWQS, we generated chemical concentration data over several different exposure scenarios, where the scenarios varied in the number of chemical groups, the amount of correlation among the chemicals, and the strength of group associations with the outcome. There were three sets of scenarios (Scenarios A–C), where the scenarios differed according to the number of total chemicals and number of chemicals in each group. Each scenario had a range of true exposure effect strengths (Strengths 1–5), starting with a null effect and odds ratio (OR) = 1.00 and then increasing in strength (both positive and negative associations) for each group. For positive associations, strengths 2–5 represent odds ratios of 1.50, 2.00, 2.50, and 3.00 respectively. Negative associations were the reciprocals of the aforementioned odds ratios (0.67, 0.50, 0.40, 0.33, respectively). Within each scenario and exposure strength, three strengths of correlation amongst the chemical concentrations were considered: (1) weak correlation of 0.1 across group and 0.5 within group (W), (2) moderate correlation of 0.3 across group and 0.7 within group (M), and (3) strong correlation of 0.5 across group and 0.9 within group (S). The different correlation structures were specified through a matrix and then converted into a covariance matrix. A mean vector and standard deviation vector were selected to generate the covariance matrix and hence allow construction of the data that was distributed as multivariate normal. The sample size for each scenario was 1000 observations.

In the first set of scenarios (Scenario A), two chemical groups consisting of nine different chemicals (with five in the first group of chemicals and four in the second group of chemicals) were generated. After the null effects scenario, the first group of chemicals was set to have a negative association with the outcome, while the second group of chemicals had a positive association. In each of these groups, two of the chemicals were made to be important while the remaining were set to be unimportant through the true chemical weights. The chemicals labeled unimportant were assigned a true weight of 0 while the important chemicals in each group were given an equal weight such that the sum of the weights in each group would equal 1 (e.g., weight of 0.5 for each of two important chemicals in a group).

In the second set of scenarios (Scenario B), a total of 14 chemicals were divided into three groups with group 1 containing the first five, group 2 the next four, and group 3 the last five. After a null effects scenario, group 1 had a negative association with the outcome while groups 2 and 3 had a positive association. Only one chemical per group was important as specified through the weights. The third set of scenarios (Scenario C) was a slight modification of Scenario B where this time groups 1 and 3 had three important chemicals while group 2 had two. The different simulation scenario terms are listed in Table 1 as a reference. These terms are used when presenting the results of the simulation study.

After defining the exposure scenarios, we created binary outcomes for case or control status to replicate a case-control study by having a relatively balanced number of cases and controls (50% ± 10% cases) in each iteration of data generation. Each data set generated was split in half to form a training dataset and a testing dataset (500 observations each). The binary outcome y was distributed as y~Binomial(n,p) where $p\;=\;\frac{1}{1+e^{\eta }}$ and $\eta \;=\;\beta_{0}^{*}\;+\;\sum_{j=1}^{K}\beta_{j}^{*}[WQS_{j}^{*}{}_{\;}]\;$, where the star notation indicates true parameter values. As no covariates were used in generation of the data, the term $z^{T}\phi \;=\;0$. The number of quantiles used in all simulations was set at four when computing the weighted index for each group (i.e., $q_{ij}=0,1,2,3$). Each scenario was simulated with 100 data sets.

For a comparison of GWQS with other methods, we also fitted WQS regression, lasso [29], and group lasso [30] to the simulated data using quantiles of exposures. Lasso imposes an L1 penalty (tuned by the parameter λ) to a traditional regression model that can shrink some regression coefficients to zero while satisfying an objective function (i.e., minimizing AIC for model fit):(3)$${\hat{\beta }}_{lasso}=\underset{\beta }{\arg\min}\left[\sum_{i=1}^{n}{\left(y_{i}-\beta_{0}-\sum_{j=1}^{p-1}\beta_{j}x_{ij}\right)}^{2}+\lambda \sum_{j=1}^{p}\left|\beta_{j}\right|\right].$$

The group lasso applies a penalty to groups of the predictors as:(4)$${\hat{\beta }}_{group}=\underset{\beta }{\arg\min}\left[\sum_{i=1}^{n}{\left(y_{i}-\beta_{0}-\sum_{j=1}^{p-1}\beta_{j}x_{ij}\right)}^{2}+\sum_{j}\lambda_{j}\left\|\beta_{j}\right\|\right].$$

For both lasso and group lasso, we used 10-fold cross-validation for choosing the penalty term that minimized the Akaike Information Criterion (AIC). AIC is an estimator of out-of-sample prediction error, often used to compare the fit of different models to the same data (smaller AIC is better). We used a bi-level (group and component level) penalty known as group minimax concave penalty (MCP) for the group lasso to be more in spirit with the dimension reduction in the GWQS model [31]. More specifically, the group MCP places an outer MCP penalty on a sum of inner MCP penalties for each group, resulting in differential shrinkage of regression coefficients for components inside a group (in contrast to uniform shrinkage of all components in a group).

To assess the performance of the models, we calculated the mean squared error (MSE), bias, and power on each of the group exposure effects and the sensitivity and specificity of identifying chemicals as important or not. Power is the probability of a hypothesis test detecting an effect if there is an effect to be found (i.e., chemical group associated with childhood leukemia). When calculating power, we used α = 0.05 to determine the significance of the association of each chemical group with the outcome. We measured sensitivity by determining the proportion of important chemicals that were identified by the models as being important. This was done by determining if the estimated weight of the important chemicals produced by the GWQS and WQS models was greater than or equal to the threshold $\frac{1}{C_{j}}$. Likewise, we defined specificity as the proportion of the unimportant chemicals that were correctly deemed unimportant by the GWQS and WQS models. This was determined by checking if the estimated weights of the unimportant chemicals were less than the same threshold of $\frac{1}{C_{j}}$.

For lasso and group lasso, the threshold on the estimated regression coefficients for calculating sensitivity and specificity was 0.0001. As p-values are typically not computed for these lasso methods, we calculated power in two different ways and presented them as a lower and upper bound. The lower bound was calculated by placing all the chemicals with non-zero weights into a generalized linear model and determining whether any chemicals in each group had a p-value less than 0.05. The proportion of datasets where this occurred was calculated as the lower bound of power. The upper bound was calculated by determining the proportion of datasets where at least one chemical in group j had an estimated effect size greater than or equal to $\frac{1}{C_{j}}$. Because the lasso models do not provide a group exposure effect coefficient like GWQS and WQS, we calculated MSE and bias using the differences of the true group effect multiplied by the true weight and the estimated chemical regression coefficients from the model. We summed the estimated regression coefficients overall for lasso and by group for the group lasso and exponentiated to get estimated health effects from the lasso and group lasso models.

### 2.3. Data Analysis

As an application of the grouped index method to actual data, we used GWQS regression to analyze childhood leukemia in the CCLS. The CCLS is a population-based case-control study conducted in northern and central California (17 counties in the San Francisco Bay area and 18 counties in the Central Valley) designed to examine the relationships between various environmental exposures, genetic factors, and childhood leukemia [17]. Cases ≤ 14 years of age were identified within 72 h after diagnosis from the nine major pediatric clinical centers in the study area from 1995 to 2012. Eligible criteria include (1) age under 15 years, (2) without prior cancer diagnosis, (3) residence in California at the time of diagnosis, and (4) having an English or Spanish-speaking biological parent. Controls were selected from California birth certificate files and matched to cases on date of birth, sex, Hispanic ethnicity, and maternal race.

As part of a second interview (first interviews were conducted from December 1999 to November 2007) dust samples were collected from homes of cases and controls younger than 8 years at diagnosis (similar reference date for controls) who were still living at the diagnosis home. Eligibility was limited to younger cases and controls so that the carpet dust sample would reflect exposures over a substantial portion of the child’s early life. A total of 277 eligible cases and 308 eligible controls participated in the second interview (*n* = 583).

Dust samples were collected by high-volume small surface sampler (HVS3) from a carpet or rug in the room where the child spent the most time while awake (commonly the family room). Vacuum dust samples were collected by removing the used bag or by emptying the loose dust from the household vacuum cleaner into a sealable polyethylene bag. The household vacuum was found to be a reasonable alternative to the HVS3 for detecting, ranking, and quantifying the concentrations of pesticides and other compounds [32]. As previously described, concentrations of 64 organic chemicals were measured using gas chromatography/mass spectrometry (GC/MS) in multiple ion monitoring mode after extraction with three different extraction methods [32]. Nine metals were measured using microwave-assisted acid digestion combined with inductively coupled plasma/mass spectrometry (ICP/MS). Due to missing covariate information for some participants, 268 cases and 296 controls were included in this analysis (*n* = 564). We considered exposure to 49 chemicals (Appendix A) for which at least 20% of the measurements were above the limit of detection. Chemicals concentrations below the limit of detection were imputed to be between 0 and the detection limit using univariate imputation assuming a lognormal distribution.

The concentrations of some of the chemicals measured in the house dust samples were strongly correlated for some chemical pairs. Chemicals with the strongest correlation were in the same chemical class. For example, several PAHs were highly correlated (*r* > 0.6) with each other (e.g., *r* = 0.90 for benzo[a]anthracene, chrysene). Some chemicals or congeners within the following chemical classes were also highly correlated: organochlorine insecticides, pyrethroid insecticides, and PCBs. The strong correlations between chemicals in these chemical classes prohibits using traditional regression methods to model cumulative chemical exposure effects and requires mixture analysis methods.

To analyze the association of exposure to the 49 chemicals and childhood leukemia, we placed the chemicals into the following six chemical groups: PCBs, PAHs, insecticides, herbicides, metals, and the tobacco exposure markers of nicotine and cotinine. These groupings were based on structural similarity (e.g., PCBs, PAH) or their use (herbicides, insecticides). We placed the fungicide ortho-phenylphenol in the herbicide group. For the analysis, we estimated the exposure effect for each of the six groups simultaneously for childhood leukemia using GWQS regression. In addition to modeling groups of environmental chemical exposures, we adjusted for the following covariates: child’s age, sex, ethnicity, annual household income, mother’s education level, mother’s age at birth of child, at whether the child lived at the sampling residence since birth. In fitting the GWQS model, we used four quantiles with 100 bootstraps and a 50–50 split of training and validation datasets We summarized the results using ORs for each group along with 95% confidence intervals and forest plots. We also assessed the important chemical exposures in the group using the estimated weights.

## 3. Results

### 3.1. Simulation Study

We present effect estimate and power results for scenarios A and B in the main text, while these results for scenario C can be found in the Appendix A. Table 2 shows the estimated odds ratios for each of the models in scenario sets A. It is clear that GWQS estimated the odds ratios well in both scenario sets, but lasso and WQS regression performed much worse. The lasso and WQS regression do not distinguish the group effects and so the estimates in scenario set A approximate the average effect of the groups. This causes the lasso and WQS effect estimates in scenario set A to be around an odds ratio of 1 or less (Table 2). The group lasso estimates in scenario set A are more in agreement with those from GWQS and are very close to the truth. In scenario B, the odds ratio estimates from WQS and the lasso are larger than 1 because there are two positively associated groups and only one negatively associated group (Table 3). In this scenario, WQS overestimates the true odds ratio for any one group (except for the null effect case). Again, the GWQS and group lasso estimates are closer to the truth than the lasso estimates. While the GWQS model had a tendency to overestimate the strength of association in this scenario set, the group lasso shrank the estimates closer to the true value. Also notable is that the GWQS effect estimates were closer to the truth when the strength of correlation in exposures was strongest. This is an important result because traditional regression methods are most challenged to estimate exposure effects when the correlation among exposures is high. The results for scenario set C were similar to scenario set B (see Appendix A).

The power values in Table 4 for scenario set A and Table 5 for scenario set B reveal some differences in power and type I error (rate of false positives in significance testing) across the methods. For scenario A, GWQS had considerably higher power than WQS regression, which was also underpowered compared to the group lasso and lasso. In scenario B, the power values were very similar across the methods, likely due to the increased signal in three groups over the two in scenario set A. For type I error, the values were similar across the four models when considering the lower bound for lasso and the group lasso. However, the null effect scenario shows that the type I error rate is high for the upper bound in both lasso and group lasso in Table 4 and Table 5. The type I error rate between GWQS and WQS were similar in all conditions and scenarios. Increases in the correlation in exposures seemed to have a negligible effect on power.

The sensitivity (Table 6) and specificity (Table 7) of the four models across scenario sets A–C show some patterns in model performance. Generally, as the correlation structure of the chemical concentration data becomes stronger, the sensitivity and specificity both decrease. This implies that it is harder to distinguish the important from unimportant chemicals as the correlation between all chemicals increases. As expected, the sensitivity and specificity increased with the strength of association. Comparing index approaches, the sensitivity and specificity are higher for GWQS than WQS across scenarios. The lasso and group lasso had high sensitivity, but relatively low specificity compared with GWQS and WQS. This means that lasso and group lasso both identified too many chemicals as being important when compared with the truth. This finding for the lasso is consistent with earlier work [23], but the group lasso finding is new in the context of mixture analysis. Comparing results between pairs of scenario sets reveals that the sensitivity and specificity increased when there were two important chemicals in each group (scenario C) compared with one important chemical per group (scenario B) and decreased when going from two groups (scenario A) to three groups (scenario B) in the mixture. This implies that model performance could decrease with increasing complexity of the mixture in terms of the number of groups, but this decrease could be offset by an increase in the number of important members in each group.

Finally, Table 8 compares the AIC between the four models, and it is clear that the model fit is substantially worse for lasso and group lasso compared with GWQS and WQS regression. The large AIC for the lasso and group lasso relative to GWQS and WQS is likely due to excessive shrinking of individual chemical components. While the AIC values are similar for lasso and group lasso, they are consistently smaller for the group lasso, and the difference in favor of the group lasso increases as the number of groups increases. AIC values are similar between GWQS and WQS regression for the null effect, but the difference between them becomes more apparent as the magnitudes of association increase, where the AIC is lower with GWQS. Thus, the results demonstrate that GWQS had the best model fit compared to the other three models throughout all conditions and scenarios.

### 3.2. Application to Childhood Leukemia

The simulation study results above demonstrate that GWQS regression is the best method considered for modeling exposure effects for several groups of chemicals, and we have used it to model risk of childhood leukemia in the CCLS. The results from estimating exposure effects for six chemical groups while adjusting for the covariates described above are in Table 9. Herbicides have a statistically significant positive effect (OR = 1.791, *p* = 0.021) while insecticides have a significant negative effect (OR = 0.434, *p* = 0.010). PCBs and PAHs have marginally significant positive effects. The marginal significance is clear from the forest plot of estimates in Figure 1. From the figure, the chemical groups with almost no evidence of an association with childhood leukemia include metals, and tobacco. Variability appears to be similar for all groups with the exception of herbicides, which has a wide confidence interval relative to the other chemical groups. For the covariates, those children living in the highest income households ($75,000 or more) had significantly reduced odds (OR = 0.292, *p*-value = 0.036) of having leukemia, and having lived in the sampling household since birth was marginally significant (OR = 0.608, *p*-value = 0.075).

The estimated weights for the chemicals in significant or marginally significant groups from the GWQS model are shows in Figure 2. Amongst the herbicides, dacthal is the important chemical with a weight of 0.313. Among the marginally significant groups of PCBs and PAHs, PCB 138 and PAH indeno-123cd-pyrene are the most important chemicals, with weights of nearly 0.521 and 0.263, respectively. For insecticides, the most important chemical was carbaryl, albeit with an inverse association with leukemia.

## 4. Discussion

In this paper, we evaluated the proposed method of grouped weighted quantile sum regression for estimating exposure effects for many groups of compounds with different directions of association with the outcome, as well as identifying the important components in each group. We compared this method with the existing approaches of WQS regression, lasso, and the group lasso. There were several notable findings from the simulation study. First, the simulation study demonstrated the limitation of WQS regression and the lasso to estimate the exposure effects for more complicated mixtures with positive and negative association groups. Both methods produced an effect estimate that was an averaging of positive and negative effects, with lasso shrinking the estimate toward the null. Second, GWQS and the group lasso produced effect estimates that were very close to the truth. Third, GWQS had better power to detect true exposure effects compared with WQS. Fourth, GWQS had better sensitivity and specificity than WQS for identifying important chemicals. Fifth, both GWQS and WQS had better specificity than lasso and group lasso. Finally, GWQS had the best model goodness-of-fit compared to WQS, lasso, and group lasso.

These results are encouraging for the use of GWQS in applied studies. The result of increased specificity compared with lasso and group lasso is in agreement with a previous result for WQS regression and lasso that showed improved specificity for WQS compared with lasso and the elastic net [23]. While the lasso and group lasso shrink coefficients, they tend to not shrink the regression coefficients to zero for enough of the unimportant chemicals. This results in a high sensitivity but reduced specificity. Effectively, the shrinkage methods indicate too many exposures as being important (i.e., false positives). Comparing the performance of GWQS to WQS, it is easy to conclude that GWQS should be used instead of WQS whenever there are at least two natural groupings of exposures. This could occur with different chemical classes, or with a positive association group and negative association group. WQS will underestimate the true positive association group effect by including negative association components into the single index. This will result in biased regression coefficients, reduced power, and reduced sensitivity. Currently, GWQS requires that the chemical groups be specified in advance, as is also true for the group lasso.

In the application of GWQS to the study of childhood leukemia in California, we found a significant positive association with herbicides and a significant negative association with insecticides. The most important herbicide was dacthal, while the most important insecticide was carbaryl. Both PCBs and PAHs had marginally significant positive associations with childhood leukemia, with PCB 138 and indeno-123cd-pyrene standing out as the most important PCB and PAH, respectively. There was little evidence for association with metals and tobacco markers. These findings add to the existing literature on environmental chemicals exposures and childhood leukemia risk. While we adjusted for many covariates in the analysis, it is impossible to rule out residual confounding as a factor when interpreting the findings.

The herbicide dacthal (also known as chlorthal) can be considered as being associated with the greatest increase in risk for childhood leukemia in this study. Dacthal is used as an herbicide for over 30 years in California, particularly for broccoli, and may remain in the soil for up to three months and leak into the groundwater [33]. The EPA classifies dacthal as Group 3, which is not classified as a carcinogen [34]. Only one previous CCLS analysis examined the association of dacthal and childhood leukemia. That study conducted univariable logistic regression with individual chemicals and found a positive but not statistically significant association between quantiles of dacthal concentrations and childhood acute lymphocytic leukemia (ALL) risk [33]. There was, however, a significant elevated risk of ALL associated with the presence of dacthal in the dust (detected vs. not detected OR = 1.52, 95% CI:1.03, 2.23) [33]. Our finding adds significant evidence of a positive association with dacthal and childhood leukemia.

The PAH indeno-123cd-pyrene has previously been found to be positively associated with ALL in the CCLS, with an OR = 1.81 (95% CI: 1.04, 3.16) in the subset of children with a vacuum dust sample (*n* = 160) after using multiple imputation of chemical concentrations below the limit of detection [35]. The EPA classifies indenopyrene as a probable human carcinogen (Group 2B) [36].

Our findings suggest an association between PCB 138 and childhood leukemia, which adds to the evidence from a previous analysis of PCBs and organochlorine pesticides in the CCLS (using case and control participants with HVS3 dust samples only). Ward et al. [11] found significant positive trends in ALL risk using univariable logistic regression with increasing concentrations of PCB congener 138, as well as congeners 118 and 153. PCBs are known carcinogens for humans, specifically for melanoma, with limited evidence for breast cancer and non-Hodgkin lymphoma, [37], and were banned for manufacturing in the United States by the EPA in 1977.

The significant negative associations we observed for insecticides are consistent with results from univariable and multivariable logistic regression that included carbaryl, chlorpyrifos, and diazinon [38]. Our results are also somewhat consistent with previous cancer studies in adults. Carbaryl was found to have a significant inverse association (OR = 0.5, 95%: 0.3–0.9) in the highest exposure group for NHL [8]. Other insecticides with inverse associations with NHL in this study included chlorpyrifos and diazinon. In a study of carbaryl exposure in the Agricultural Health Study, carbaryl was found to have an inverse association with prostate cancer and no association with cancer overall [39].

## 5. Conclusions

In summary, through our evaluation of GWQS it appears this method can make a substantial contribution as a statistical approach in the field of environmental epidemiology. This method considers multiple diverse environmental chemical exposures with different magnitudes and directions of associations, and allows for a more comprehensive assessment of environmental exposures. It also compares favorably with some of the existing methods used in mixture analysis. While we applied it for an environmental chemical risk analysis of childhood leukemia that considered different chemical classes, the approach will be applicable to many other diseases with suspected environmental causes. Hopefully, this approach will help facilitate and encourage future studies to uncover multiple environmental determinants of cancer.

## Figures and Tables

**Figure 1 ijerph-18-00504-f001:**
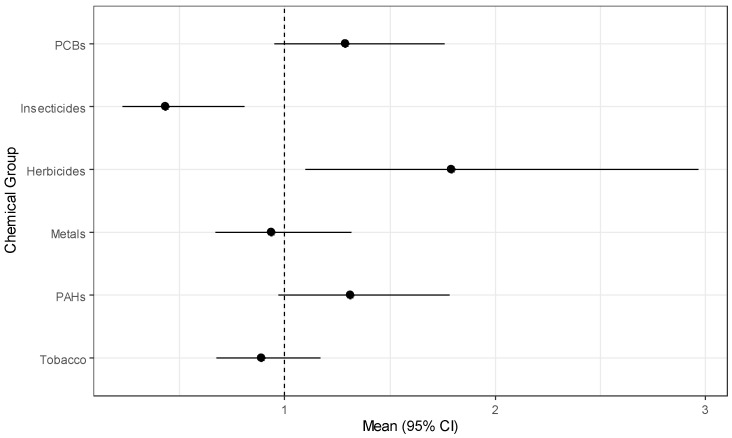
Forest plot of chemical group effects for childhood leukemia.

**Figure 2 ijerph-18-00504-f002:**
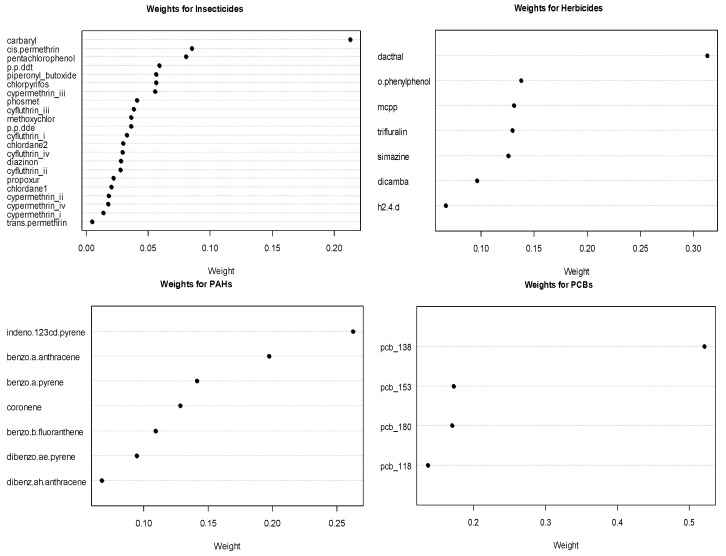
Estimated chemical weights for significant or marginally significant chemical groups.

**Table 1 ijerph-18-00504-t001:** Definition of simulation study exposure scenario terms.

Term	Definition
A	Scenario A: 9 chemicals in 2 groups (5,4); 2 important in each group
B	Scenario B: 14 chemicals in 3 groups (5,4,5); 1 important in each group
C	Scenario C: 14 chemicals in 3 groups (5,4,5); (3,2,3) important per group
1	Strength Level 1: OR=1.00 for all groups (Null effect scenario)
2	Strength Level 2: OR = (0.67, 1.50) for A; OR = (0.67, 1.50, 1.50) for B and C
3	Strength Level 3: OR = (0.50, 2.00) for A; OR = (0.50, 2.00, 2.00) for B and C
4	Strength Level 4: OR = (0.40, 2.50) for A; OR = (0.40, 2.50, 2.50) for B and C
5	Strength Level 5: OR = (0.33, 3.00) for A; OR = (0.33, 3.00, 3.00) for B and C
W	Weak Correlation Structure (0.1 across group, 0.5 within group)
M	Moderate Correlation Structure (0.3 across group, 0.7 within group)
S	Strong Correlation Structure (0.5 across group, 0.9 within group)

**Table 2 ijerph-18-00504-t002:** True and estimated odds ratios for the four models (averaged over 100 replicates) for scenario A with two chemical groups and true exposure effects listed in the Parameter column.

Parameter	GWQS	WQS	Group Lasso	Lasso
Weak Correlation				
exp (β_1_) = 1.00	1.03	1.04	1.00	1.01
exp (β_2_) = 1.00	1.01		1.00	
exp (β_1_) = 0.67	0.67	0.99	0.69	1.00
exp (β_2_) = 1.50	1.50		1.47	
exp (β_1_) = 0.50	0.50	0.97	0.51	1.01
exp (β_2_) = 2.00	2.04		1.98	
exp (β_1_) = 0.40	0.39	0.91	0.40	1.00
exp (β_2_) = 2.50	2.57		2.51	
exp (β_1_) = 0.33	0.32	0.90	0.34	1.01
exp (β_2_) = 3.00	3.11		3.04	
Moderate Correlation				
exp (β_1_) = 1.00	1.02	1.03	1.01	1.01
exp (β_2_) = 1.00	1.01		1.01	
exp (β_1_) = 0.67	0.68	1.01	0.69	1.01
exp (β_2_) = 1.50	1.51		1.48	
exp (β_1_) = 0.50	0.51	0.95	0.51	1.01
exp (β_2_) = 2.00	2.01		2.00	
exp (β_1_) = 0.40	0.40	0.90	0.40	1.01
exp (β_2_) = 2.50	2.52		2.53	
exp (β_1_) = 0.33	0.34	0.84	0.34	1.01
exp (β_2_) = 3.00	3.02		3.04	
Strong Correlation				
exp (β_1_) = 1.00	1.01	1.02	1.00	1.01
exp (β_2_) = 1.00	1.01		1.00	
exp (β_1_) = 0.67	0.68	1.00	0.68	1.01
exp (β_2_) = 1.50	1.50		1.49	
exp (β_1_) = 0.50	0.51	0.96	0.51	1.00
exp (β_2_) = 2.00	1.97		1.98	
exp (β_1_) = 0.40	0.41	0.91	0.40	1.00
exp (β_2_) = 2.50	2.48		2.52	
exp (β_1_) = 0.33	0.34	0.88	0.33	1.00
exp (β_2_) = 3.00	2.99		3.04	

**Table 3 ijerph-18-00504-t003:** True and estimated odds ratios for the four models (averaged over 100 replicates) for scenario B with three chemical groups and the true exposure effects listed in the Parameter column.

Parameter	GWQS	WQS	Group Lasso	Lasso
Weak Correlation				
exp (β_1_) = 1.00	1.00	0.99	0.99	0.99
exp (β_2_) = 1.00	0.99		1.00	
exp (β_3_) = 1.00	1.01		1.01	
exp (β_1_) = 0.67	0.61	2.10	0.67	1.45
exp (β_2_) = 1.50	1.60		1.50	
exp (β_3_) = 1.50	1.62		1.50	
exp (β_1_) = 0.50	0.45	3.39	0.50	1.92
exp (β_2_) = 2.00	2.16		2.01	
exp (β_3_) = 2.00	2.21		2.01	
exp (β_1_) = 0.40	0.36	4.45	0.40	2.40
exp (β_2_) = 2.50	2.71		2.52	
exp (β_3_) = 2.50	2.79		2.54	
exp (β_1_) = 0.33	0.30	5.35	0.33	2.86
exp (β_2_) = 3.00	3.26		3.05	
exp (β_3_) = 3.00	3.36		3.06	
Moderate Correlation				
exp (β_1_) = 1.00	1.00	1.00	0.99	0.99
exp (β_2_) = 1.00	0.99		0.99	
exp (β_3_) = 1.00	1.01			
exp (β_1_) = 0.67	0.64	1.84	0.66	1.47
exp (β_2_) = 1.50	1.55		1.51	
exp (β_3_) = 1.50	1.55		1.50	
exp (β_1_) = 0.50	0.48	2.70	0.50	1.95
exp (β_2_) = 2.00	2.06		2.00	
exp (β_3_) = 2.00	2.09		2.00	
exp (β_1_) = 0.40	0.38	3.44	0.40	2.43
exp (β_2_) = 2.50	2.57		2.51	
exp (β_3_) = 2.50	2.63		2.53	
exp (β_1_) = 0.33	0.32	4.12	0.33	2.90
exp (β_2_) = 3.00	3.08		3.06	
exp (β_3_) = 3.00	3.16		3.06	
Strong Correlation				
exp (β_1_) = 1.00	1.00	0.99	0.99	0.99
exp (β_2_) = 1.00	0.99		0.99	
exp (β_3_) = 1.00	1.01		1.00	
exp (β_1_) = 0.67	0.66	1.69	0.66	1.47
exp (β_2_) = 1.50	1.52		1.49	
exp (β_3_) = 1.50	1.52		1.51	
exp (β_1_) = 0.50	0.49	2.40	0.50	1.96
exp (β_2_) = 2.00	2.03		2.00	
exp (β_3_) = 2.00	2.03		2.04	
exp (β_1_) = 0.40	0.40	3.03	0.40	2.45
exp (β_2_) = 2.50	2.51		2.49	
exp (β_3_) = 2.50	2.54		2.54	
exp (β_1_) = 0.33	0.34	3.60	0.33	2.93
exp (β_2_) = 3.00	2.99		3.01	
exp (β_3_) = 3.00	3.02		3.05	

**Table 4 ijerph-18-00504-t004:** Power and type I error for the four models for scenario A.

Parameter	GWQS	WQS	Group Lasso	Lasso
Weak Correlation				
exp (β_1_) = 1.00	0.00	0.03	(0.11, 0.11)	(0.19, 0.03)
exp (β_2_) = 1.00	0.03		(0.06, 0.06)	
exp (β_1_) = 0.67	0.91	0.43	(0.47, 0.98)	(1.00, 0.99)
exp (β_2_) = 1.50	0.98		(0.48, 0.99)	
exp (β_1_) = 0.50	1.00	0.67	(0.51, 1.00)	(1.00, 1.00)
exp (β_2_) = 2.00	1.00		(0.51, 1.00)	
exp (β_1_) = 0.40	1.00	0.77	(0.48, 1.00)	(1.00, 1.00)
exp (β_2_) = 2.50	1.00		(0.48, 1.00)	
exp (β_1_) = 0.33	1.00	0.78	(0.53, 1.00)	(1.00, 1.00)
exp (β_2_) = 3.00	1.00		(0.53, 1.00)	
Moderate Correlation				
exp (β_1_) = 1.00	0.03	0.02	(0.11, 0.22)	(0.19, 0.10)
exp (β_2_) = 1.00	0.04		(0.07, 0.13)	
exp (β_1_) = 0.67	0.94	0.40	(0.45, 0.99)	(0.95, 0.99)
exp (β_2_) = 1.50	1.00		(0.49, 1.00)	
exp (β_1_) = 0.50	1.00	0.67	(0.51, 1.00)	(1.00, 1.00)
exp (β_2_) = 2.00	1.00		(0.51, 1.00)	
exp (β_1_) = 0.40	1.00	0.71	(0.55, 1.00)	(1.00, 1.00)
exp (β_2_) = 2.50	1.00		(0.55, 1.00)	
exp (β_1_) = 0.33	1.00	0.74	(0.52, 1.00)	(1.00, 1.00)
exp (β_2_) = 3.00	1.00		(0.52, 1.00)	
Strong Correlation				
exp (β_1_) = 1.00	0.05	0.03	(0.09, 0.20)	(0.17, 0.17)
exp (β_2_) = 1.00	0.04		(0.05, 0.15)	
exp (β_1_) = 0.67	0.94	0.30	(0.40, 0.99)	(0.76, 0.98)
exp (β_2_) = 1.50	1.00		(0.40, 0.99)	
exp (β_1_) = 0.50	1.00	0.43	(0.50, 1.00)	(0.97, 1.00)
exp (β_2_) = 2.00	1.00		(0.48, 1.00)	
exp (β_1_) = 0.40	1.00	0.57	(0.51, 1.00)	(1.00, 1.00)
exp (β_2_) = 2.50	1.00		(0.50, 1.00)	
exp (β_1_) = 0.33	1.00	0.60	(0.51, 1.00)	(1.00, 1.00)
exp (β_2_) = 3.00	1.00		(0.51, 1.00)	

**Table 5 ijerph-18-00504-t005:** Power and type I error for the four models for scenario B.

Parameter	GWQS	WQS	Group Lasso	Lasso
Weak Correlation				
exp (β_1_) = 1.00	0.09	0.07	(0.02, 0.00)	(0.34, 0.00)
exp (β_2_) = 1.00	0.09		(0.07, 0.00)	
exp (β_3_) = 1.00	0.05		(0.07, 0.00)	
exp (β_1_) = 0.67	0.99	0.97	(1.00, 1.00)	(1.00, 1.00)
exp (β_2_) = 1.50	1.00		(1.00, 0.98)	
exp (β_3_) = 1.50	0.98		(1.00, 0.99)	
exp (β_1_) = 0.50	1.00	1.00	(1.00, 1.00)	(1.00, 1.00)
exp (β_2_) = 2.00	1.00		(1.00, 1.00)	
exp (β_3_) = 2.00	1.00		(1.00, 1.00)	
exp (β_1_) = 0.40	1.00	1.00	(1.00, 1.00)	(1.00, 1.00)
exp (β_2_) = 2.50	1.00		(1.00, 1.00)	
exp (β_3_) = 2.50	1.00		(1.00, 1.00)	
exp (β_1_) = 0.33	1.00	1.00	(1.00, 1.00)	(1.00, 1.00)
exp (β_2_) = 3.00	1.00		(1.00, 1.00)	
exp (β_3_) = 3.00	1.00		(1.00, 1.00)	
Moderate Correlation				
exp (β_1_) = 1.00	0.09	0.07	(0.05, 0.02)	(0.27, 0.00)
exp (β_2_) = 1.00	0.05		(0.04, 0.00)	
exp (β_3_) = 1.00	0.06		(0.02, 0.01)	
exp (β_1_) = 0.67	0.99	1.00	(1.00, 1.00)	(1.00, 1.00)
exp (β_2_) = 1.50	0.99		(0.99, 0.98)	
exp (β_3_) = 1.50	0.97		(1.00, 1.00)	
exp (β_1_) = 0.50	1.00	1.00	(1.00, 1.00)	(1.00, 1.00)
exp (β_2_) = 2.00	1.00		(1.00, 1.00)	
exp (β_3_) = 2.00	1.00		(1.00, 1.00)	
exp (β_1_) = 0.40	1.00	1.00	(1.00, 1.00)	(1.00, 1.00)
exp (β_2_) = 2.50	1.00		(1.00, 1.00)	
exp (β_3_) = 2.50	1.00		(1.00, 1.00)	
exp (β_1_) = 0.33	1.00	1.00	(1.00, 1.00)	(1.00, 1.00)
exp (β_2_) = 3.00	1.00		(1.00, 1.00)	
exp (β_3_) = 3.00	1.00		(1.00, 1.00)	
Strong Correlation				
exp (β_1_) = 1.00	0.09	0.07	(0.04, 0.11)	(0.23, 0.05)
exp (β_2_) = 1.00	0.07		(0.05, 0.05)	
exp (β_3_) = 1.00	0.06		(0.05, 0.18)	
exp (β_1_) = 0.67	0.97	1.00	(0.87, 0.99)	(1.00, 1.00)
exp (β_2_) = 1.50	0.95		(0.94, 0.94)	
exp (β_3_) = 1.50	0.98		(0.93, 1.00)	
exp (β_1_) = 0.50	1.00	1.00	(1.00, 1.00)	(1.00, 1.00)
exp (β_2_) = 2.00	1.00		(1.00, 1.00)	
exp (β_3_) = 2.00	1.00		(1.00, 1.00)	
exp (β_1_) = 0.40	1.00	1.00	(1.00, 1.00)	(1.00, 1.00)
exp (β_2_) = 2.50	1.00		(1.00, 1.00)	
exp (β_3_) = 2.50	1.00		(1.00, 1.00)	
exp (β_1_) = 0.33	1.00	1.00	(1.00, 1.00)	(1.00, 1.00)
exp (β_2_) = 3.00	1.00		(1.00, 1.00)	
exp (β_3_) = 3.00	1.00		(1.00, 1.00)	

**Table 6 ijerph-18-00504-t006:** Sensitivity for the four models for scenarios A–C.

Correlation-Effect	GWQS	WQS	Group Lasso	Lasso
Scenario A				
Weak-1	0.38	0.36	0.13	0.08
Moderate-1	0.34	0.32	0.12	0.11
Strong-1	0.34	0.32	0.09	0.10
Weak-2	0.85	0.61	0.99	1.00
Moderate-2	0.75	0.57	0.90	0.97
Strong-2	0.64	0.52	0.67	0.89
Weak-3	0.97	0.74	1.00	1.00
Moderate-3	0.88	0.62	0.99	1.00
Strong-3	0.77	0.60	0.91	0.99
Weak-4	1.00	0.74	1.00	1.00
Moderate-4	0.82	0.69	1.00	1.00
Strong-4	0.94	0.63	0.99	1.00
Weak-5	1.00	0.74	1.00	1.00
Moderate-5	0.97	0.69	1.00	1.00
Strong-5	0.87	0.65	1.00	1.00
Scenario B				
Weak-1	0.39	0.36	0.21	0.07
Moderate-1	0.38	0.33	0.12	0.05
Strong-1	0.40	0.34	0.12	0.08
Weak-2	0.69	0.51	1.00	1.00
Moderate-2	0.58	0.43	1.00	1.00
Strong-2	0.51	0.41	0.97	0.99
Weak-3	0.81	0.56	1.00	1.00
Moderate-3	0.70	0.50	1.00	1.00
Strong-3	0.60	0.44	1.00	1.00
Weak-4	0.88	0.58	1.00	1.00
Moderate-4	0.78	0.54	1.00	1.00
Strong-4	0.65	0.47	1.00	1.00
Weak-5	0.92	0.60	1.00	1.00
Moderate-5	0.80	0.55	1.00	1.00
Strong-5	0.70	0.50	1.00	1.00
Scenario C				
Weak-1	0.35	0.37	0.20	0.10
Moderate-1	0.36	0.34	0.13	0.06
Strong-1	0.41	0.35	0.12	0.08
Weak-2	1.00	0.73	0.89	0.96
Moderate-2	0.99	0.66	0.73	0.92
Strong-2	0.90	0.61	0.51	0.81
Weak-3	1.00	0.69	0.99	1.00
Moderate-3	1.00	0.67	0.93	0.99
Strong-3	0.98	0.65	0.73	0.94
Weak-4	1.00	0.67	1.00	1.00
Moderate-4	1.00	0.67	0.96	1.00
Strong-4	0.99	0.66	0.84	1.00
Weak-5	1.00	0.67	1.00	1.00
Moderate-5	1.00	0.67	0.99	1.00
Strong-5	1.00	0.67	0.89	1.00

**Table 7 ijerph-18-00504-t007:** Specificity for the four models for scenarios A–C.

Correlation-Effect	GWQS	WQS	Group Lasso	Lasso
Scenario A				
Weak-1	0.56	0.62	0.87	0.91
Moderate-1	0.59	0.67	0.87	0.87
Strong-1	0.62	0.65	0.89	0.86
Weak-2	0.91	0.88	0.74	0.54
Moderate-2	0.84	0.87	0.78	0.56
Strong-2	0.78	0.78	0.81	0.55
Weak-3	0.97	0.95	0.76	0.52
Moderate-3	0.92	0.92	0.80	0.52
Strong-3	0.84	0.86	0.81	0.51
Weak-4	0.99	0.97	0.73	0.47
Moderate-4	0.96	0.95	0.80	0.55
Strong-4	0.91	0.90	0.80	0.52
Weak-5	1.00	0.97	0.78	0.46
Moderate-5	0.97	0.96	0.81	0.49
Strong-5	0.91	0.92	0.83	0.50
Scenario B				
Weak-1	0.58	0.65	0.79	0.89
Moderate-1	0.60	0.67	0.87	0.92
Strong-1	0.62	0.68	0.86	0.90
Weak-2	0.87	0.86	0.67	0.64
Moderate-2	0.80	0.81	0.80	0.64
Strong-2	0.76	0.78	0.84	0.64
Weak-3	0.96	0.92	0.50	0.60
Moderate-3	0.88	0.87	0.83	0.59
Strong-3	0.82	0.83	0.85	0.62
Weak-4	0.99	0.97	0.40	0.55
Moderate-4	0.93	0.90	0.83	0.54
Strong-4	0.86	0.86	0.83	0.60
Weak-5	1.00	0.97	0.33	0.56
Moderate-5	0.95	0.92	0.83	0.55
Strong-5	0.90	0.89	0.84	0.60
Scenario C				
Weak-1	0.58	0.65	0.79	0.89
Moderate-1	0.60	0.67	0.87	0.92
Strong-1	0.61	0.68	0.85	0.90
Weak-2	0.91	0.90	0.77	0.50
Moderate-2	0.86	0.87	0.79	0.52
Strong-2	0.79	0.82	0.81	0.53
Weak-3	0.98	0.95	0.79	0.49
Moderate-3	0.94	0.94	0.81	0.45
Strong-3	0.87	0.88	0.81	0.50
Weak-4	0.99	0.97	0.78	0.42
Moderate-4	0.97	0.97	0.79	0.44
Strong-4	0.91	0.92	0.83	0.52
Weak-5	0.99	0.98	0.78	0.39
Moderate-5	0.99	0.97	0.82	0.44
Strong-5	0.94	0.93	0.81	0.49

**Table 8 ijerph-18-00504-t008:** AIC between the four models for scenarios A–C (lower is better).

Correlation-Effect	GWQS	WQS	Group Lasso	Lasso
Scenario A				
Weak-1	691.55	690.39	1373.49	1373.80
Moderate-1	691.55	690.41	1373.44	1373.69
Strong-1	691.48	690.43	1373.67	1373.77
Weak-2	670.55	687.64	1328.32	1329.62
Moderate-2	668.88	688.03	1325.85	1326.96
Strong-2	670.66	688.82	1331.70	1333.27
Weak-3	632.01	681.78	1247.89	1248.99
Moderate-3	630.61	682.89	1246.08	1246.84
Strong-3	638.61	687.06	1265.08	1265.93
Weak-4	594.42	675.44	1173.30	1174.47
Moderate-4	593.96	679.37	1171.27	1172.02
Strong-4	606.39	684.61	1198.35	1198.98
Weak-5	563.46	670.63	1111.38	1112.81
Moderate-5	564.74	675.55	1110.28	1111.08
Strong-5	579.61	683.29	1142.86	1143.80
Scenario B				
Weak-1	691.35	689.57	1373.35	1374.72
Moderate-1	691.17	689.50	1374.20	1374.89
Strong-1	691.07	689.43	1374.01	1374.63
Weak-2	668.24	678.21	1265.23	1269.12
Moderate-2	659.44	670.90	1276.92	1280.47
Strong-2	657.29	668.09	1292.50	1295.69
Weak-3	619.02	644.76	1107.18	1111.40
Moderate-3	603.42	634.65	1131.83	1135.89
Strong-3	597.94	628.59	1156.26	1159.62
Weak-4	573.47	613.91	984.68	989.29
Moderate-4	554.41	603.05	1014.54	1018.92
Strong-4	547.97	594.91	1047.73	1051.08
Weak-5	537.53	590.68	894.71	899.49
Moderate-5	514.27	577.45	923.04	927.43
Strong-5	508.31	568.59	959.91	963.60
Scenario C				
Weak-1	691.35	689.57	1373.35	1374.72
Moderate-1	691.17	689.50	1374.20	1374.89
Strong-1	691.07	689.43	1374.01	1374.63
Weak-2	642.22	667.67	1322.34	1326.40
Moderate-2	647.62	665.49	1306.67	1310.72
Strong-2	653.84	667.48	1304.53	1309.04
Weak-3	564.53	616.52	1219.96	1222.62
Moderate-3	578.33	620.81	1189.60	1192.36
Strong-3	589.46	624.55	1183.72	1186.90
Weak-4	506.15	580.72	1130.46	1122.18
Moderate-4	522.14	587.31	1090.23	1092.56
Strong-4	537.60	590.31	1080.11	1082.51
Weak-5	462.98	555.70	1055.30	1058.11
Moderate-5	479.31	561.11	1006.79	1008.93
Strong-5	496.39	562.43	996.39	998.65

**Table 9 ijerph-18-00504-t009:** GWQS regression odds ratio estimates for chemical groups and demographic covariates.

Variable	Odds Ratio	2.5% CI	97.5% CI	*p*-Value
Chemical Groups				
PCBs	1.29	0.95	1.76	0.11
Insecticides	**0.43**	**0.23**	**0.81**	**0.01**
Herbicides	**1.79**	**1.10**	**2.97**	**0.02**
Metals	0.94	0.67	1.32	0.71
PAHs	1.31	0.97	1.78	0.08
Tobacco	0.89	0.68	1.17	0.41
Child’s age	1.00	0.86	1.16	0.99
Female	1.05	0.62	1.76	0.86
Child’s Ethnicity				
Hispanic	0.97	0.47	1.96	0.92
Non-Hispanic	1.52	0.77	3.00	0.23
Household Income				
$15,000–$29,999	0.88	0.28	2.68	0.82
$30,000–$44,999	0.80	0.25	2.48	0.70
$45,000–$59,999	0.53	0.15	1.76	0.30
$60,000–$74,999	0.54	0.15	1.91	0.35
$75,000 or more	**0.29**	**0.09**	**0.90**	**0.04**
Mother’s education				
High school	0.99	0.28	3.46	0.99
Some college	1.25	0.34	4.59	0.73
Bachelor’s or higher	0.99	0.26	3.72	0.99
Mother’s age	1.01	0.96	1.07	0.65
Residence since birth	0.61	0.35	1.05	0.08

## Data Availability

The CCLS data presented in this study are available on request from the senior author. The data are not publicly available due to privacy restrictions.

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
