# Peer review of "Assessment of Grouped Weighted Quantile Sum Regression for Modeling Chemical Mixtures and Cancer Risk"

_ijerph, 2021, doi:10.3390/ijerph18020504_

Round 1

Reviewer 1 Report

General comments.

Interesting paper on the issue of unpicking the effects of mixtures and health effects, and presents a new method to account for exposure mixtures being chemical contaminants.

The paper proposes a new methodology, and then using this method in a real world example to demonstrate its use. This would be of interest to the readership of the journal so I would recommend publication in this journal.

I have a few minor comments to address on the text below.

For the real world example of disentangling the effects of chemical mixtures on childhood leukaemia’s, I have more comments on the design of the study and accuracy of estimating exposures.

Are the right exposure metrics measured? For the outcome variable – childhood leukaemia- what are the likely or suspect environmental exposures? Have these been measured? Have all potential confounders been taken account of? A mixed effects was found with the different chemical groups, I suspect as the proxies used for exposures, and indeed environmental sampling method used, may not be the best for measuring or estimating exposures. If these aspects have not been measured or accounted for, then they need to be stated as limitations ( indeed there is not a limitations section of the paper).

Is your R package you designed available for public use? Can you reference it in an online repository (Gitlab etc?)?

Are references in correct format for journal? Do they have to be numbered?

Specific comments on the text.

Abstract: Line26 – what is lasso? Needs a brief explanation for the uninitiated.

Line 40 – you could refer to the EU REACH classification of chemicals for another comprehensive list.

Line 78- can you briefly explain what WQS is or does briefly here?

Line 101- define lasso and/or explain what it is here.

Line 146- is your R package group WQS available on a GitHub/GitLab for others to use it? If so please reference.

Paragraph 223- what is the time period for the cohort (Time Place Person) can you describe the epidemiology/case definition?

Line 235 – why was vacuum samples taken to represent household exposures, as opposed to other methods, e.g. dust wipe samples or air sampling? How were the samples taken, and how long was left between vacuuming?

Para 256. What other confounders could/should be measured? E.g. Parents occupation, especially if in a manual/industrial profession that is exposed to chemicals which can contaminate the household? Distance to an industrial plant/heavy industry zone? Was time lived in residence measured? How was ‘Whether child lived at sampling residence since birth’ recorded? Was this a binary outcome? Length of time lived would be a better measure for proxy of exposure.

If these parameters cannot be defined or were not measured then they need to be stated as limitations of the study.

Fig 2- Correct y axis labels to fit page.

Line 390- do these associations make sense? Is this explained by what is known in exposure science and toxicology? What are the mechanisms of effect?

Line 423: Can you reference an already published paper where this association was found?

What is the evidence for the overall associations found with these chemicals, in the literature? Have there been any meta-analysis published?

Author Response

General comments.

Interesting paper on the issue of unpicking the effects of mixtures and health effects, and presents a new method to account for exposure mixtures being chemical contaminants.

The paper proposes a new methodology, and then using this method in a real world example to demonstrate its use. This would be of interest to the readership of the journal so I would recommend publication in this journal.

RESPONSE: We thank the reviewer for the positive feedback.

I have a few minor comments to address on the text below.

For the real world example of disentangling the effects of chemical mixtures on childhood leukaemia’s, I have more comments on the design of the study and accuracy of estimating exposures.

Are the right exposure metrics measured? For the outcome variable – childhood leukaemia- what are the likely or suspect environmental exposures? Have these been measured? Have all potential confounders been taken account of? A mixed effects was found with the different chemical groups, I suspect as the proxies used for exposures, and indeed environmental sampling method used, may not be the best for measuring or estimating exposures. If these aspects have not been measured or accounted for, then they need to be stated as limitations ( indeed there is not a limitations section of the paper).

RESPONSE: The objective of the California Childhood Leukemia Study (CCLS) was to evaluate potential environmental risk factors for childhood leukemia. To accomplish this, the study investigators attempted to measure a large number (~100) of chemicals in house dust for cases and controls in the study. These chemical concentrations measured in the home are the main environmental exposures of concern. In fact, it is unusual for an epidemiologic study of cancer to measure such a large number of exposures inside the home as was done in the CCLS. Regarding confounders, we adjusted for all potential confounders available in the CCLS, including age, sex, ethnicity, household income, mother’s education, and if the dust sample home is the residence since birth. While a large number of exposures and potential confounders were considered in this analysis, not all possible relevant exposures or potential confounders could be included. In other words, residual confounding is always possible. We have added this as a limitation of our analysis with the following text (line 422): “While we adjusted for many covariates in the analysis, it is impossible to rule out residual confounding as a factor when interpreting the findings.”

Is your R package you designed available for public use? Can you reference it in an online repository (Gitlab etc?)?

RESPONSE: As stated in the paper, our R package is available for public use. The reference for the package available on CRAN is included in the paper and is:

Wheeler D. & Carli M. groupWQS: Grouped Weighted Quantile Sum Regression. 2020; R package version 0.0.3.

Are references in correct format for journal? Do they have to be numbered?

RESPONSE: To follow the IJERPH format for publication, references must be numbered in order of appearance in the text. We have changed the references and citations to match this format.

 Specific comments on the text.

Abstract: Line26 – what is lasso? Needs a brief explanation for the uninitiated.

RESPONSE: We have spelled out the acronym for lasso as the least absolute shrinkage and selection operator in the Abstract.

Line 40 – you could refer to the EU REACH classification of chemicals for another comprehensive list.

RESPONSE: We appreciate the suggestion, but believe that our citation is adequate in this case.

Line 78- can you briefly explain what WQS is or does briefly here?

RESPONSE: We have modified the text (starting on line 79) to be more descriptive of WQS regression.

“Weighted quantile sum (WQS) regression [18, 23-24] is a constrained regression approach that was designed to estimate the effect of a mixture of correlated chemicals and identify the individual chemicals most strongly associated with a health outcome while adjusting for risk factors. In WQS, a weight is estimated for each chemical in a weighted index, where the weights are constrained to be between 0 and 1 and sum to 1.”

Line 101- define lasso and/or explain what it is here.

RESPONSE: We have defined the lasso as the least absolute shrinkage and selection operator here and added more description for the lasso with the text below (starting line 104). We also added an equation for the lasso in section 2.2.

“We also compared the performance of GWQS regression with WQS regression, the least absolute shrinkage and selection operator (lasso), and the group lasso in estimating exposure effects and identifying important chemicals. Shrinkage methods such as the lasso have been used previously to model chemical mixture effects because they were designed for handling correlated predictor variables [23-24].”

Line 146- is your R package group WQS available on a GitHub/GitLab for others to use it? If so please reference.

RESPONSE: We have modified the text to make it clear that our R package is publicly available on CRAN.

Paragraph 223- what is the time period for the cohort (Time Place Person) can you describe the epidemiology/case definition?

RESPONSE: Participants were enrolled in the CCLS from 1995 to 2012. We have added this detail in section 2.3 and have modified the text to have more details about the CCLS (line 239):

“The CCLS is a population-based case-control study conducted in northern and central California (17 counties in the San Francisco Bay area and 18 counties in the Central Valley) designed to examine the relationships between various environmental exposures, genetic factors, and childhood leukemia [17]. Cases ≤ 14 years of age were identified within 72 hours after diagnosis from the nine major pediatric clinical centers in the study area from 1995 to 2012. Eligible criteria include (1) age under 15 years, (2) without prior cancer diagnosis, (3) residence in California at the time of diagnosis, and (4) having an English or Spanish-speaking biological parent. Controls were selected from California birth certificate files and matched to cases on date of birth, sex, Hispanic ethnicity, and maternal race.”

Line 235 – why was vacuum samples taken to represent household exposures, as opposed to other methods, e.g. dust wipe samples or air sampling? How were the samples taken, and how long was left between vacuuming?

RESPONSE: Settled dust found indoors is a mixture of biologically derived materials, particles deposited from indoor aerosols, particles deposited from building materials (e.g., deteriorated paint), and soil particles that infiltrate from outdoors (e.g., soil tracked indoors on shoes). Because typical cleaning removes only a portion of dust from indoor environments (e.g., when vacuuming a carpet), indoor dust acts as a reservoir for chemical contamination. Settled dust can be an important source of chemical exposures, especially for young children, who have frequent hand-to-mouth contact.

We have modified existing text in section 2.3 to add more details about how the dust samples were taken (line 254):

“Dust samples were collected by high-volume small surface sampler (HVS3) from a carpet or rug in the room where the child spent the most time while awake (commonly the family room). Vacuum dust samples were collected by removing the used bag or by emptying the loose dust from the household vacuum cleaner into a sealable polyethylene bag.”

Participants were asked to avoid household vacuuming for a week prior to dust collection.

Para 256. What other confounders could/should be measured? E.g. Parents occupation, especially if in a manual/industrial profession that is exposed to chemicals which can contaminate the household? Distance to an industrial plant/heavy industry zone?

RESPONSE: While residual confounding is always possible, what the reviewer has described regarding parent’s occupation is not confounding. If a parent was exposed to chemicals at work and he/she tracked those chemicals back into the home where the child was exposed to them, then that activity would be “upstream” of the relationship being measured in the paper. In other words, we are asking whether the contamination in the home increased the child’s risk of leukemia. We are not trying to figure out the source of the contamination. The second question regarding source of contamination has been addressed in other papers – for example see Gunier et al: https://www.ncbi.nlm.nih.gov/pmc/articles/PMC3222988/.

Was time lived in residence measured? How was ‘Whether child lived at sampling residence since birth’ recorded? Was this a binary outcome? Length of time lived would be a better measure for proxy of exposure. If these parameters cannot be defined or were not measured then they need to be stated as limitations of the study.

RESPONSE: The CCLS does include information about the length of time each participant spent living in the residence from which the dust sample was collected. However, we disagree with the reviewer’s comment that this would be a better “proxy for exposure” because any other homes that the participant lived in would also have chemical contamination – potentially more. Rather, we consider time spent living in the dust-collection residence to be a surrogate for the precision of the exposure assessment. Stratifying the analysis using the binary variable of having lived in the sampling home would be an approach to investigate potentially different effects for those having lived in the sampling home since birth. If we look at just those participants who have been living in the same house since birth, perhaps we would see stronger effects, assuming that the in utero/very early life period is an etiologically-relevant window of exposure. However, the power of GWQS is substantially reduced by doing a stratified analysis with a small case-control study such as the CCLS. For this reason, we have included the binary variable as a covariate in the GWQS regression model. We are currently working on another method for mixture analysis that has greater power for small studies and will apply this to the CCLS in a future paper.

Fig 2- Correct y axis labels to fit page.

RESPONSE: We have resized the figure to fit the page.

Line 390- do these associations make sense? Is this explained by what is known in exposure science and toxicology? What are the mechanisms of effect?

RESPONSE: In lines 424-455 of the Discussion we place our findings in context of the literature. We believe that this is an accurate summary of the existing literature.

Line 423: Can you reference an already published paper where this association was found?

RESPONSE: The paper that we have cited is in review at Environment International and is the only other paper of which we are aware that studies the association between these particular insecticides and childhood leukemia.

What is the evidence for the overall associations found with these chemicals, in the literature? Have there been any meta-analysis published?

RESPONSE: There is no meta-analysis that we are aware of for the association between these particular insecticides and childhood leukemia. We report in the Discussion the known inverse associations between these insecticides and cancers (lines 450-455).

Reviewer 2 Report

This paper is a step ahead to grasp the effect of a multiple of chemicals on the probability of contracting cancer. The paper is written in clear language and explains the development of the statistical method well. It is recommendable to read the previous paper in which the method was developed for one group of related chemicals and then reads the present one on a multiple of groups. The structure of the paper is conventional and draws no comment. The applied testing methods and statements of confidence and significance are generally accepted. The whole makes a very professional impression.   For a general reader it may be recommendable to explain in a few words what is measured with power, type-I error, sensitivity and specificity, and Akaike's Information Criterion.   As shown by the figures results are convincing.

Author Response

This paper is a step ahead to grasp the effect of a multiple of chemicals on the probability of contracting cancer. The paper is written in clear language and explains the development of the statistical method well. It is recommendable to read the previous paper in which the method was developed for one group of related chemicals and then reads the present one on a multiple of groups. The structure of the paper is conventional and draws no comment. The applied testing methods and statements of confidence and significance are generally accepted. The whole makes a very professional impression.   For a general reader it may be recommendable to explain in a few words what is measured with power, type-I error, sensitivity and specificity, and Akaike's Information Criterion.   As shown by the figures results are convincing.

RESPONSE: We appreciate the kind words from the reviewer. We have added definitions for AIC (line 207) and power (line 216) in section 2.2 and type I error on line 319. Sensitivity (line 219) and specificity (line 222) are defined in section 2.2.

Author Response

The current study aimed to assess the performances of a relatively new methodological approach for estimating the effects of grouped and highly correlated covariates on some outcome. The manuscript is well written and potentially useful for the readers. Please find below some comments and suggestions for improving the manuscript.

General comments

Why didn’t the authors include Ridge regression as a possible alternative method in the simulation study? Indeed, Ridge regression would be more appropriate than Lasso regression for dealing with highly correlated variables (while Lasso is more appropriate for doing variable selection). As an extreme example, if you have two identical covariates in the same model, Lasso will probably set one of the two coefficients to 0 and will assign the whole importance to the other, while Ridge will distribute the importance equally between the two covariates. Ridge may therefore work better than Lasso for the scope of the paper.

RESPONSE: We selected lasso instead of ridge regression as a comparison method because we wanted to use a method that could shrink regression coefficients to zero. GWQS can shrink chemical exposure weights to zero, so lasso is a fairer comparison than ridge regression. In addition, we wanted to compare to a group shrinkage method (i.e. group lasso) because our method of GWQS is a group-based model. There is no commonly used group ridge regression model to use as a comparison. Moreover, we previously evaluated the shrinkage methods of lasso, adaptive lasso, and elastic net for modeling chemical mixtures (Carrico et al. 2014) and found that the performances of lasso and the elastic net were very similar in terms of sensitivity and specificity. Note that elastic net is the lasso with a ridge constraint added to the model.

The Authors may consider to mention possible drawbacks/limitations of GWQS regression. For example, it appears that the chemical group composition should be known/defined a priori. In other words, the model is not able to identify the groups of chemical by itself. In this regard, the author may also mention another method that may be useful for addressing the scope of GWQS regression, that is plsDA (PérezEnciso, M., & Tenenhaus, M. (2003). Prediction of clinical outcome with microarray data: a partial least squares discriminant analysis (PLS-DA) approach. Human genetics, 112(5-6), 581-592). Indeed, plsDA would also be able to give itself indications about the group composition (as the sets of variables associated with each latent component, similarly to principal component analysis).

RESPONSE: Indeed, one of the steps in using GWQS regression is grouping the chemicals into groups. However, this step is also required when using the group lasso. When using GWQS, we typically group chemicals by class/family, but other approaches are possible. We are currently working on other approaches for empirically grouping chemical exposures, including using correlations between chemicals. We have added a comment on this point in the Discussion (line 413): “Currently, GWQS requires that the chemical groups be specified in advance, as is also true for the group lasso.”

We thank the reviewer for the suggestion of PLS-DA and will also investigate this method.

There appears to be some information missing about the case in which none of the chemicals is important within a given group. What should we expect in that case? Should we expect that all the chemical in the same group will be given a weight of approximately 1/cj? What happened to the Ws in the metal and tobacco groups in the application to childhood leukemia? Does GWQS regression require the a-priori assumption that at least one chemical per group is influent?

RESPONSE: In the case that no chemicals are important within a group, the chemicals will receive approximately equal weight (1/c_j) and the estimated regression coefficient for that group would be approximately null (odds ratio of approximately 1). There is no assumption in GWQS that there must be at least one important chemical within a group.

Materials and methods

It may be worth presenting the model equation for all the models included in the simulation study, not only for GWQS regression. It would also be useful, for the other models, to highlight how the Authors derived (from the estimated model parameters) estimates for the beta j coefficients, the distributions of which are summarized in Table 2. Indeed, this is not fully clear, especially for the models without any grouping structure (WQS and Lasso). This would also clarify lines 218 to 221, which appear a bit unclear.

RESPONSE: We have added equations for the lasso and the group lasso in section 2.2. In addition, we added the following clarifying text (line 236): “We summed the estimated regression coefficients overall for lasso and by group for the group lasso and exponentiated to get estimated health effects from the lasso and group lasso models.”

In the Table 2 caption, “True and estimated odds ratios for the four models...” would not be fully correct, a more correct version could be: “True and estimated odds ratios (averaged over 1,000 replicates) for the four models...”.

RESPONSE: We have added text to the table captions to indicate that the odds ratios are averaged over 100 data sets.

Line 124: please clarify that the quantiles qij will assume increasing integer values, such as 1, 2, 3 and 4 for quartiles or 1, 2, 3, ..., 10 for deciles. It will become clear later at lines 191, but it would be worth clarifying here too.

RESPONSE: We have clarified the text to define quantiles for example as quartiles 0, 1, 2, 3 (line 124).

Line 130: “For estimation of the model parameters in equation (1)...”. Actually, it appears that the estimation process involves two steps: the Ws are estimated in the first step, while the beta j are estimated in the second step. However, it also seems that the beta j are also estimated for each bootstrap sample, even if they are temporarily ignored (we will only retain the final estimates obtained from equation 2). If so, aren’t there identifiability concerns for the betas and the Ws when estimating equation 1? How are they worked around?

RESPONSE: The constraint on the weights helps with model identifiability. The results of the simulation study (Table 2-7) demonstrate that GWQS is able to identify both the regression coefficients and the important chemicals with good power and sensitivity.

Lines 193-194. It is not clear how 10-fold CV and AIC criteria were combined for choosing the penalty term, since they are usually used as surrogate criteria.

RESPONSE: The AIC is used as the performance metric, but applies to one set of data only. Cross-validation (e.g., 5- or 10-fold) is used to get a better evaluation of model performance by using several sets of data. This avoids overfitting to one data set, allowing the model to perform better in prediction when new/different data sets are used.